# Incidence of Orbital Side Effects in Zygomaticomaxillary Complex and Isolated Orbital Walls Fractures: A Retrospective Study in South Italy and a Brief Review of the Literature

**DOI:** 10.3390/jcm12030845

**Published:** 2023-01-20

**Authors:** Umberto Committeri, Antonio Arena, Emanuele Carraturo, Martina Austoni, Cristiana Germano, Giovanni Salzano, Giacomo De Riu, Francesco Giovacchini, Fabio Maglitto, Vincenzo Abbate, Paola Bonavolontà, Luigi Califano, Pasquale Piombino

**Affiliations:** 1Maxillofacial Surgery Unit, Department of Neurosciences, Reproductive and Odontostomatological Sciences, University of Naples “Federico II”, 80131 Naples, Italy; 2Otolaryngology and Maxillo-Facial Surgery Unit, Istituto Nazionale Tumori—IRCCS Fondazione G. Pascale, 80131 Naples, Italy; 3Maxillofacial Surgery Operative Unit, University Hospital of Sassari, 07100 Sassari, Italy; 4Maxillofacial Surgery Unit, Santa Maria Della Misericordia Hospital, San Sisto, 06121 Perugia, Italy

**Keywords:** diplopia, zygomaticomaxillary complex fractures, isolated orbital walls fractures, orbit, enophtalmos, exophtalmos

## Abstract

Zygomaticomaxillary complex and isolated orbital walls fractures are one of the most common fractures of the midface, often presenting orbital symptoms and complications. Our study was born with the aim of understanding the trend in the incidence of orbital presurgical symptoms, specifically diplopia, enophthalmos and exophthalmos, in the Campania Region in southern Italy. We conducted a retrospective, monocentric observational study at the Maxillofacial Surgery Unit of the Federico II University Hospital of Naples, enrolling 402 patients who reported a fracture of the zygomaticomaxillary complex and orbital floor region from 15 June 2021 to 15 June 2022. Patients were evaluated by age, gender, etiology, type of fracture, preoperative orbital side effects and symptoms. Pre-surgical side effects were studied, and 16% of patients (*n* = 66) developed diplopia. Diplopia was most common in patients previously operated on for orbital wall fractures (100%), and least common in patients who reported trauma after interpersonal violence (15%) and road traffic accidents (11%). Exophthalmos appeared only in 1% (six cases); whereas it did not appear in 99% (396 cases). Enophthalmos was present in 4% (sixteen cases), most commonly in interpersonal violence cases (two cases). The frequency of orbital complications in patients with zygomaticomaxillary complex and isolated orbital walls fractures suggests how diplopia remains the most common pre-surgical orbital side effect.

## 1. Introduction

In the field of cranio-maxillofacial traumatology, zygomaticomaxillary complex and isolated orbital walls fractures are one of the most common fractures of the midface, and account for around 27% of all facial fractures, being second only to nasal fractures [1]. Injury patterns may be isolated to the orbit or form part of a much larger zygomatic-maxillary complex (ZMC) or pan-facial fracture patterns. These fractures are usually classified as pure and impure. Impure orbital fractures are those that involve the orbital rim(s) with the internal orbit walls. Most orbital pure fractures occur along the floor and/or medial walls of the orbit, where the walls are the thinnest [2,3]. Orbital wall fractures are also classified as isolated fractures, involving a single orbital wall, or as combined fractures, when more than one orbital wall is affected [4]. Following the anatomical region of the fracture rim, orbital fractures can be divided into orbital floor, orbital roof, median and lateral wall fractures; the floor is the most frequently injured because it contains the largest open space and lacks support. The frequency of orbital fractures has become more common owing to the increasing amount of traffic accidents, industrial accidents, sport-related injuries and physical assaults, and, rarely, gunshots [5,6,7]. Management and treatment of orbital fractures poses a challenge to every surgeon and physician in general. This is because of their complex anatomy and their innate relationship to relevant structures, such as the globe, optic nerve and ophthalmic artery, among others, and their direct influence on the most precious of senses: vision. For this reason, they represent the few real urgencies in the realm of Cranio-Maxillofacial trauma [8]. Orbital symptoms are a relatively common complication of orbital fractures. In the medical literature, they occur in about 20% of patients, most frequently in the subgroup of orbital blow fractures. Diagnosis is essential to permit early treatment, as various symptoms, such as diplopia and enophthalmos, may persist even after surgical treatment, especially if the diagnosis is delayed [9,10]. This epidemiological study was born with the aim of understanding the trend in the incidence of orbital presurgical symptoms, specifically diplopia, enophthalmos and exophthalmos, in the Campania Region in southern Italy to help the development of a trauma patient protocol based on the clinical presentation and specific demands.

## 2. Materials and Methods

The study was conducted at the Maxillofacial Surgery Unit of the Federico II University Hospital, Regional Referral Center for Cranio-Maxillo-Facial Traumas (Article 1 paragraph 203 L.F. Campania Region 2011, Italy). It was a retrospective, monocentric observational study. The sample size consisted of 402 patients who reported a fracture of the zygomaticomaxillary complex and orbital floor. All clinical investigations and procedures were conducted according to the principles expressed in the Declaration of Helsinki. Ethical approval to access and use the data was obtained from the Federico II/Cardarelli Research Ethic Committee (81/2022). Patients were evaluated by age, gender, etiology, type of fracture, preoperative orbital side effects and symptoms. Blind/visually impaired patients, patients suffering from osteoporosis/osteomalacia or psychiatric disorders were excluded. Patients admitted in this study did not suffer from any globe injuries, the management of which would take priority over any maxillofacial procedures. All data were extrapolated from medical records from 15 June 2021 to 15 June 2022. Definitive diagnosis was obtained by performing computed tomography (CT) (Figure 1).

Preoperative side effects (diplopia, enophtalmos, exophthalmos) were evaluated through accurate ophthalmic clinical examinations by the same operator and using the same instrument (Hertel Exophthalmometer, © 2022 Lombart Instrument, Inc. All Rights Reserved. 800-LOMBART, 5358 Robin Hood Road, Norfolk, Virginia 23513). In the same way, all patients took the Hess-Lancaster screen or Hess-Lancaster red-green test to better diagnose any defect in ocular motility. The considered variables were summarized considering frequency and percentage for each category.

## 3. Results

The most affected patients were men, representing 77% of the total number of cases (out of a sample of 308 men and 94 women). The most represented age group was between 13–25 years with 22% (90 cases), followed by the 25–37 age group with 19% (76 cases), the 37–49 age group (76 cases) with 19%, the 49–61 age group with 16% (64 cases), the 61–73 age group with 12% (48 cases), the 73–85 age group with 8% (34 cases), and, finally, the 85–97 age group with 3% (14 cases). The mean age reported was 46 years.

From the data obtained, we analyzed the mechanism of the trauma that caused the fracture. In our series, the most frequent causes were accidental falls in 37% (148 cases), road accidents in 36% (144 cases) and interpersonal violence in 16% (66 cases). Among the minor causes, on the other hand, syncopal episodes were found in 4% (sixteen cases), sports accidents in 2% (ten cases), previous fracture outcomes in 3% of cases (fourteen cases), and, finally, 1% were workplace accidents (four cases). From the analyzed data, we noticed that mostly women reported accidental falls, in 60% of cases, and reported road accidents in 30%. Men mostly reported having road accidents (37%), then accidental falls (30%), followed by interpersonal violence (21%).

The fractures were stratified according to the presence or absence of the involvement of the orbital frame, as impure in 64% (258 cases) and in pure in 36% (144 cases), with a frequency ratio of 1.8:1.

These fractures were then further classified according to the area of impact (Figure 2):

Zygomatic orbital fractures, the most frequent, in 60% (242 cases)Fractures of the orbital floor in 26% (104 cases)Fractures of the medial wall in 9% (38 cases)Fractures of the lateral wall in 3% (12 cases)Fractures of the orbital roof in 1% (6 cases)

Among the orbital side effects, we found the following (Figure 3):

Diplopia: 84% of patients did not report double vision (336 cases), 16% reported diplopia (66 cases).

Exophthalmos appeared only in 1% (6 cases), while it did not appear in 99% (396 cases).

Enophthalmos was present in 4% (16 cases), while it was not present in 96% (386 cases).

Correlating the type of fracture with the orbital side effects (Table 1), we found, that among orbital floor fractures, 42% of patients reported diplopia (*n* = 44); in 17% of lateral wall fractures diplopia (*n* = 2) was reported, in 16% of medial wall fractures diplopia (*n* = 6) was reported, and in 6% of zygomaticomaxillary fractures diplopia (*n* = 14) was reported.

Considering exophthalmos, it was found in 2% of orbital floor fractures (*n* = 2) and in 2% of zygomaticomaxillary fractures (*n* = 4).

Enophthalmos was reported in 10% of orbital floor fractures (*n* = 10), and in 11% of medial orbital wall fractures (*n* = 4).

Correlating the cause of the trauma and the orbital complications (Figure 4), it was found that, in the studied cohort, 100% of diplopia cases were present in patients who had previously undergone surgery for orbital wall fractures (*n* = 14), 40% while practicing sports (*n* = 4), 15% occurred in interpersonal violence cases (*n* = 10), 14% in accidental fall cases (*n* = 20), 13% in syncopal episode cases (*n* = 10) and 11% in road traffic accident (*n* = 16), while none occurred in a workplace setting.

Exophthalmos in the considered cohort (Figure 5) were present in 8% of syncopal episodes cases (*n* = 2), 3% in interpersonal violence cases (*n* = 2) and in 1.3% of road traffic accidents (*n* = 2).

Enophthalmos (Figure 6) were present in 42% of patients who had previously undergone surgery for orbital wall fractures (*n* = 6), 12% in interpersonal violence cases (*n* = 8) and in 1.3% of accidental falls (*n* = 2).

## 4. Discussion

Fractures of the orbitozygomaticomaxillary complex are among the most common fractures of the midface and account for approximately 27% of all facial fractures. Impure orbital fractures are more common than pure orbital fractures [1,2,11].

Isolated orbital wall fractures account for 4% to 16% [3,12].

Early recognition of ocular injuries is fundamental in mid-facial fracture cases. The management of globe injuries often takes precedence over the treatment of mid-facial and orbital fractures. Every surgeon who addresses orbital trauma must consider how to handle an emergency surgery, whereas the fracture pattern leads to optical nerve damage and then vision loss [13].

Diagnosis is essential to permit early treatment, as various symptoms, such as diplopia and enophthalmos, may persist even after surgical treatment, especially if the diagnosis is delayed [4,14]. It is always necessary to ascertain the mechanism that caused the lesion and to reconstruct the patient’s medical history before performing a clinical examination of the orbit and globe. The initial ophthalmological assessment should include periorbital examination, visual acuity, ocular motility, pupillary responses, visual fields and a fundoscopic examination [14].

Exophthalmometry is used to measure the position of the globe, while graphic radiographic visualization with coronal CT makes it possible to detail soft tissue not visible with conventional X-rays; Coronal CT scans of 1.5 to 3 mm visualize antral soft tissue densities, such as prolapsed orbital fat, extraocular muscle and hematoma [15].

Diplopia is one of the most common post-traumatic symptoms of orbital fractures [14]. Post-traumatic monocular diplopia can be caused by extrusion of the extraocular muscles or orbital soft tissue, injury to the extraocular muscles, edema of the infraorbital adipose tissue or vertical deviation of the eyeball [16]. Any change in orbital volume directly impacts the position of the globe and its anteroposterior projection and super-inferior position. Enophthalmos can be defined as the displacement of the eyeball in a posterior direction and is attributed to an increase in the intra-orbital volume, while the term exophthalmos refers to a forward displacement of the eyeball [17].

The aim of the study conducted was to show the epidemiological distribution of pre-operative orbital symptoms (diplopia, enophtalmos and exophthalmos) presented by patients who suffered from zygomaticomaxillary complex or isolated orbital wall fractures enrolled in the Oral and Maxillofacial Operative Unit of AOU “Federico II”.

The results found in this study agree with the literature, as zygomaticomaxillary complex and isolated orbital walls fractures are the most frequent midface maxillofacial trauma [8,18]. Indeed, 60% (*n* = 242) of the 402 patients involved in this study had a zygomaticomaxillary complex fracture.

The male/female ratio was 3,4:1, with male involvement representing 77% of total cases, with a mean age reported at 46 years. The fractures were stratified according to the presence or absence of the engagement of the orbital frame, as impure in 64% (258 cases) and as pure in 36% (144 cases), with a frequency ratio of 1.8:1; therefore, the incidence is comparable to other clinical studies [11].

In our series, the most common cause was represented by accidental falls, with 148 cases (37% of the total), 92 of which were men and 56 were women. From the analyzed data, we noticed that mostly women reported accidental falls, with 60% among female cases, and then road accidents with 30%. Men mostly reported having road accidents (37%), accidental falls (30%), followed by interpersonal violence (21%). Among the minor causes, on the other hand, syncopal episodes were found in 4% of cases, sports accidents in 2%, previous fracture outcomes in 3% of cases and, finally, 1% were workplace accidents. This result may be related to a greater tendency in the Campania region to use private means of transport compared to public transport. This trend has been found in similar studies conducted in other Western countries, where it is common for a family to own at least one motor vehicle [19].

In our series, we noted sixty-six patients suffering from pre-operative diplopia (16% of the sample), six patients affected with exophthalmos (1% of the sample) and sixteen patients suffering from enophthalmos (4% of the sample). These findings are not completely supported by the literature, as Bartoli et al. reported diplopia to be a main preoperative complication in 20.2% of patients, followed by enophthalmos (2.3%) and exophthalmos (1.7%), whereas Shin et al. stated diplopia was present in 42.3% of patients [18].

In our sample, 42% of patients suffering from preoperative diplopia reported an orbital floor fracture, while 6% reported diplopia suffering from zygomaticomaxillary complex fractures; 16% had medial wall fractures and 17% had lateral wall fractures. No patients with orbital roof fractures reported diplopia. These outcomes agree with most of the international literature that reported orbital floor fractures as typically coinciding with preoperative diplopia. Ramphul et al. (2017), in their review of 126 patients with orbital floor fractures, underlined that 66.6% of the total sample suffered from preoperative diplopia [20]. A study by Burm et al. included 82 cases and reported that diplopia was associated with 25% of medial wall fractures, 80% of orbital floor fractures and 80.9% of combined medial and floor fractures [21]. Higashino et al. reported 106 cases and showed that 21.4% of medial wall fractures and 23.5% of orbital floor fractures were associated with diplopia [22]. Eun et al. reported on 387 cases in which diplopia was found on physical examination prior to surgery in 22% of medial wall fractures, 78% of floor fractures and 82% of combined medial and floor fractures [23]. Tahiri et al. reported that patients with preoperative diplopia had a 9.91 times greater postoperative risk of persistent diplopia [24].

Our results indicated that 2% of patients with orbital floor fractures showed exophthalmos and another 10% of patients with orbital floor fracture displayed enophthalmos; 2% of patients with zygomatic-maxillary complex fracture exhibited exophthalmos, 1% of presented enophthalmos and no patients with orbital roof, midwall or lateral wall fractures presented exophthalmos. An interesting observation was that enophthalmos, while it was absent in patients with orbital roof and lateral wall fractures (as exophthalmos), was found in 11% of patients with midwall fractures.

The symptoms of medial orbital wall fractures are usually less severe than those of inferior wall fractures because less muscle incarceration takes place, and the bony structure is multiply overlapped [25]. Since medial orbital wall fractures are often asymptomatic, they have received less attention in the literature [26]. However, they may cause complications such as diplopia, enophthalmos and the entrapment of extraocular muscles [27]. In particular, enophthalmos may not immediately appear after the trauma because soft tissue swelling can last weeks or months [28,29]. The international literature shows how medial wall fracture is directly correlated with enophthalmos [30].

Correlating the cause of the fracture with the pre-surgical orbital symptoms, in this study, it was found that 100% of patients who previously underwent surgery for orbital wall fractures presented diplopia, while 38% reported enophthalmos, 40% of patients who experienced trauma while practicing sports reported diplopia and none reported exophthalmos or enophthalmos.

Additionally, 15% of patients who reported an interpersonal violence accident were described as having diplopia, 33% had exophthalmos and 50% had enophthalmos.

In accidental fall cases, 14% of patients described having diplopia, 13% had enophthalmos and none described suffering from exophthalmos.

Cases correlated with syncopal falls reported diplopia in 13% of cases, and 33% reported exophthalmos.

Over a period of 1 year, 402 patients who sustained a fracture in the zygomaticomaxillary complex and isolated orbital walls fractures received a full ophthalmological examination within 1 week of injury.

## 5. Conclusions

A total of 60% (*n* = 242) of patients enrolled in this study had a zygomaticomaxillary complex fracture and 26% (104 cases) had an orbital floor fracture.

Pre-surgical side effects were studied, and 16% of patients (*n* = 66) developed diplopia. Diplopia was most common in patients who had previously undergone surgery for orbital wall fractures (100%), and was least common in those who had experienced interpersonal violence (15%) and road traffic accidents (11%). Exophthalmos only appeared in 1%, whereas it did not appear in 99% of patients. Enophthalmos was present in 4% of patients, and was more common in interpersonal violence cases.

The frequency of orbital complications in patients with zygomaticomaxillary complex and isolated orbital walls fractures complex has never been assessed before in the literature, and our findings suggest that, in patients under evaluation for orbital trauma, the observation of diplopia remains the most common orbital side effect before surgery.

## Figures and Tables

**Figure 1 jcm-12-00845-f001:**
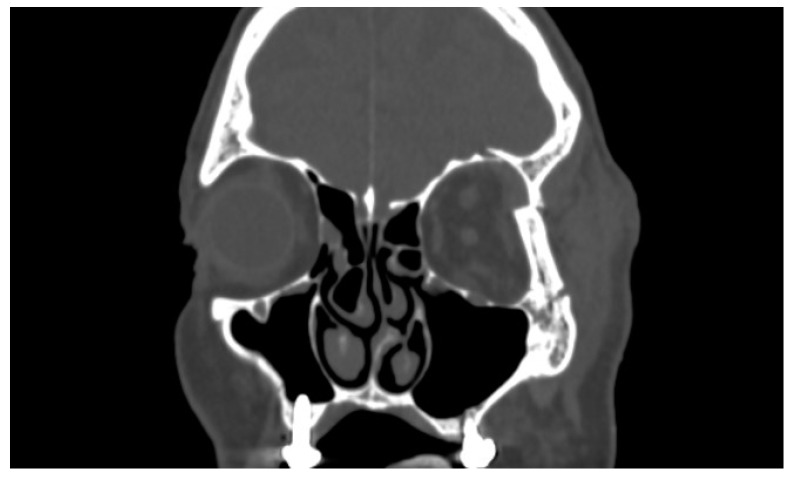
CT scan of a case of diplopia in a patient with a lateral orbit wall fracture.

**Figure 2 jcm-12-00845-f002:**
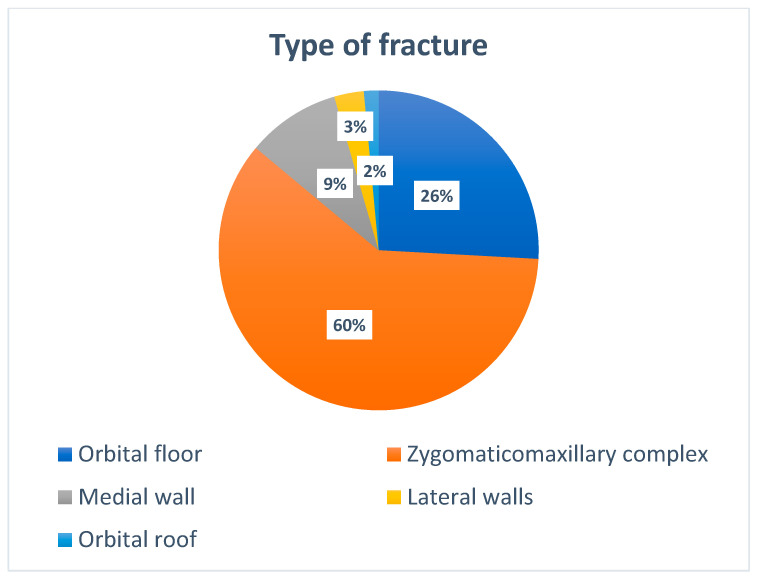
Type of fracture.

**Figure 3 jcm-12-00845-f003:**
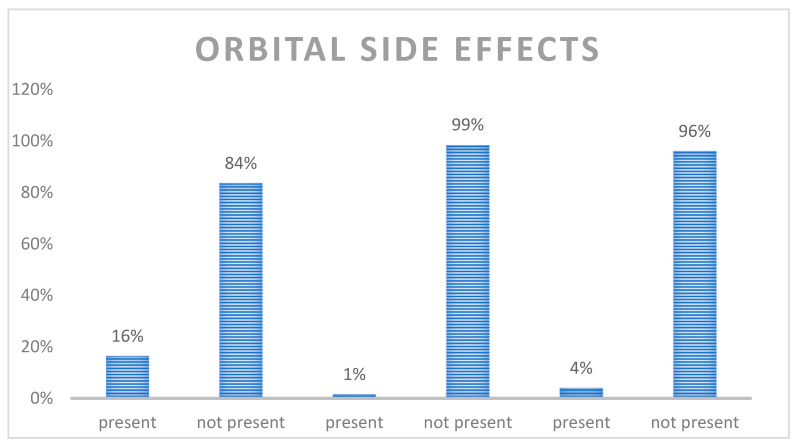
Orbital side effects.

**Figure 4 jcm-12-00845-f004:**
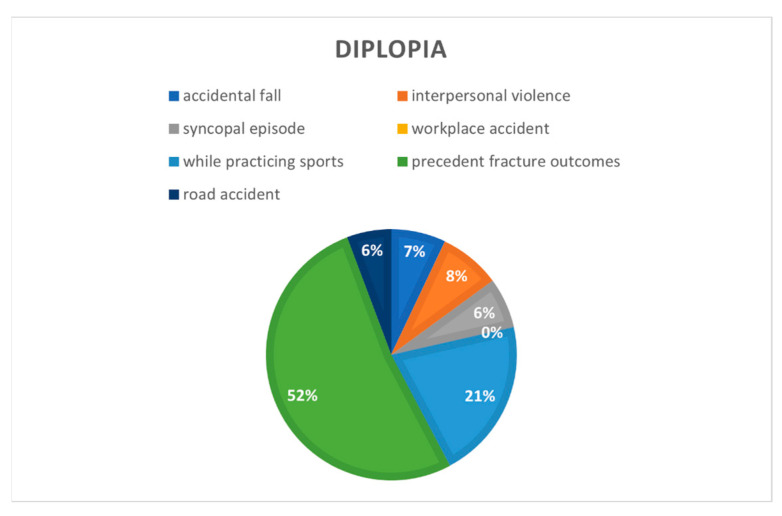
Diplopia correlated to cause of the fractures.

**Figure 5 jcm-12-00845-f005:**
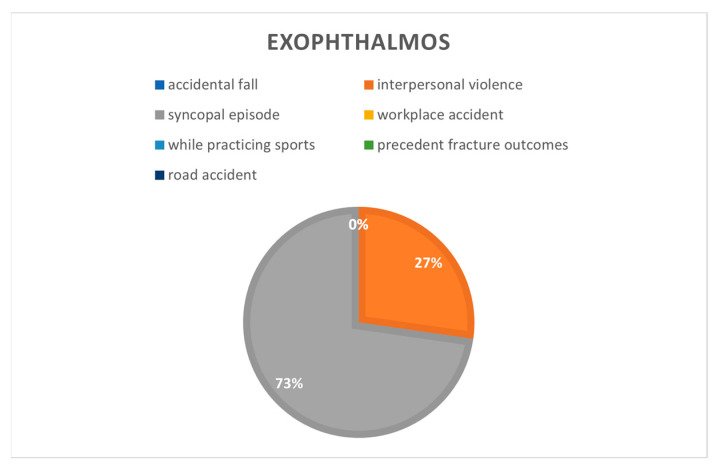
Exophthalmos correlated to cause of the fractures.

**Figure 6 jcm-12-00845-f006:**
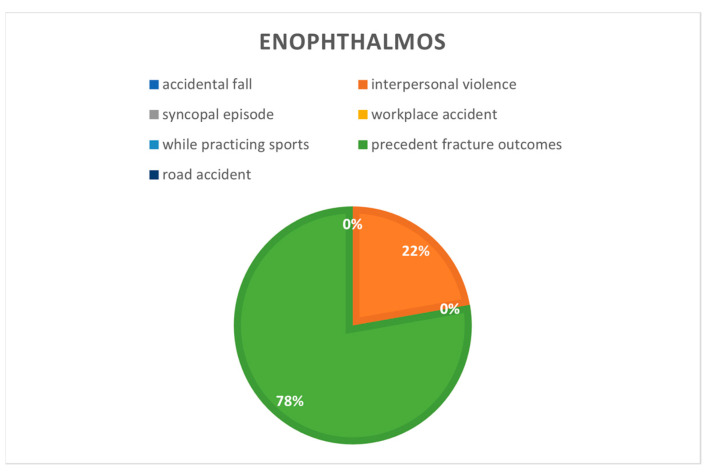
Enophthalmos correlated to cause of the fractures.

**Table 1 jcm-12-00845-t001:** Correlation between fractures and orbital side effects.

**Type of Fracture**	**Number of Fractures**	**Diplopia%**
Orbital floor	104	42%
Zygomaticomaxillary complex	242	6%
Medial wall	38	16%
Lateral walls	12	17%
Orbital roof	6	0%
**Type of Fracture**	**Number of Fractures**	**Exhophtalmos%**
Orbital floor	104	2%
Zygomaticomaxillary complex	242	2%
Medial wall	38	0%
Lateral walls	12	0%
Orbital roof	6	0%
**Type of Fracture**	**Number of Fractures**	**Enophtalmos%**
Orbital floor	104	10%
Zygomaticomaxillary complex	242	1%
Medial wall	38	11%
Lateral walls	12	0%
Orbital roof	6	0%

## Data Availability

The data are available upon request from the corresponding author.

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
