# Peer review of "Incidence of Orbital Side Effects in Zygomaticomaxillary Complex and Isolated Orbital Walls Fractures: A Retrospective Study in South Italy and a Brief Review of the Literature"

_jcm, 2023, doi:10.3390/jcm12030845_

Round 1

Reviewer 1 Report

The study provides a brief statistical description of complications related to orbital trauma in a specific region of Italy, contributing to the existing literature.

The manuscript would greatly benefit from a more organized structure and more clear terminology. Consider replacing “ocular” and ”ophthalmic”  with “orbital” (referring to complications or posttraumatic side-effects), since the terms used in the manuscript lead to confusion with actual posttraumatic injuries of the globe (globe rupture, hyphema etc.), that are also encountered in association with orbito-zygomatic fractures, but were not analysed in this study. The term orbital complications could be more suitable for the assessed variables- exophthalmos, enophthalmos and diplopia.

Additionally, it should be mentioned in the material and methods that globe injuries were not assessed, since they were not the primary focus of this study, or if they were evaluated, the data should be added to the existing results.

Consider replacing the term “precedent fracture” or better explaining the term, since it is not clear if it refers to orbital fracture sequelae due to an old untreated fracture. Also, the statistical description of precedent fractures among actual causes of fractures may not be the most suitable.

The discussions section is difficult to read and should be divided into several paragraphs.

In the first paragraph of the conclusions section (lines 266-267) please complete the sentence, since it is not understood what the total is for the percentages given. The punctuation and lines continuity should be revised.

The abbreviation CMF (line 63) should be defined.

Please rephrase lines 33-38 in the abstract, with correction of grammar for clarity.

Please rephrase lines 230-238 due to great similarity with article by Park et al. (2012) (https://www.e-aps.org/journal/view.php?doi=10.5999%2Faps.2012.39.3.204) and add article to references if information is kept.

Please rephrase lines 246-252 due to great similarity with article by Kim et al. (2017) (https://www.e-sciencecentral.org/articles/pubreader/SC000030108) and add article to references if information is kept.

Author Response

Thank you for the opportunity to revise the manuscript entitled "Incidence of orbital side effects in zygomaticomaxillary complex and isolated orbital walls fractures. A retrospective study in south Italy and a brief review of literature".

We wish to resubmit the manuscript following your suggestions.

We made an effort to revise th epoints requested and improve the manuscript accordingly.

You can find answers to yours comments attached below

Reviewer 2 Report

Paper is written to share the overall demographics, perioperative variables regarding orbital fractures. Paper is focused on a common clinical problem of orbital and periorbital fractures and complications observed with this issue. On the other hand, I strongly think that study should undergo MAJOR revision. The main points to be clarified is: 

1. Paper lack a clinical question and a strong message. It is rather just sharing some data. 

2. Orbito-zygomatic fractures is not a relevant definition. These fractures should be classified either as zygomaticomaxillary complex fractures or isolated orbital floor fractures involving either floor and/or medial wall. Most lateral wall fractures I assume are part of ZMC fractures. 

3. Are these patients operated? Are these complications observed after surgery? Most orbital floor fractures might demonstrate diplopia after a few weeks wit or without surgical intervention due to edema but presence of bone/alloplastic implant for repair or nonoperative followup might make difference from a surgical standpoint. I find it unacceptable to just say that diplopia is most common complication after these clinical condition.

4. Paper lacks clinical images, ct scans etc .when the nature of the paper is taken in consideration . 

Author Response

Thank you for the opportunity to revise the manuscript entitled "Incidence of orbital side effects in zygomaticomaxillary complex and isolated orbital walls fractures. A retrospective study in south Italy and a brief review of literature" We wish to resubmit the manuscript following your suggestions. We made an effort to revise th epoints requested and improve the manuscript accordingly. You can find answers to yours comments attached below
